# Transcriptome Profile Analysis Reveals that *CsTCP14* Induces Susceptibility to Foliage Diseases in Cucumber

**DOI:** 10.3390/ijms20102582

**Published:** 2019-05-26

**Authors:** Xuyang Zheng, Jingjing Yang, Tengxue Lou, Jian Zhang, Wenjin Yu, Changlong Wen

**Affiliations:** 1Beijing Vegetable Research Center (BVRC), Beijing Academy of Agricultural and Forestry Sciences, National Engineering Research Center for Vegetables, Beijing 100097, China; 13949140098@163.com (X.Z.); yangjingjing@nercv.org (J.Y.); loutengxue@163.com (T.L.); zhangjian@nercv.org (J.Z.); 2Beijing Key Laboratory of Vegetable Germplasms Improvement, Beijing 100097, China; 3Key Laboratory of Biology and Genetics Improvement of Horticultural Crops (North China), MOAR, Beijing 100097, China; 4Agricultural College, Guangxi University, 100 Daxue Road, Nanning 530004, Guangxi, China

**Keywords:** transcriptome, TCP, foliage diseases, leaf-specific, susceptibility

## Abstract

Foliage diseases are prevalent in cucumber production and cause serious yield reduction across the world. Identifying resistance or susceptible genes under foliage-disease stress is essential for breeding resistant varieties, of which leaf-specific expressed susceptible genes are extremely important but rarely studied in crops. This study performed an in-depth mining of public transcriptome data both in different cucumber tissues and under downy mildew (DM) inoculation, and found that the expression of leaf-specific expressed transcription factor *CsTCP14* was significantly increased after treatment with DM, as well as being upregulated under stress from another foliage disease, watermelon mosaic virus (WMV), in susceptible cucumbers. Furthermore, the Pearson correlation analysis identified genome-wide co-expressed defense genes with *CsTCP14*. A potential target *CsNBS-LRR* gene, *Csa6M344280.1*, was obtained as obviously reduced and was negatively correlated with the expression of the susceptible gene *CsTCP14*. Moreover, the interaction experiments of electrophoretic mobility shift assay (EMSA) and yeast one-hybrid assay (Y1H) were successfully executed to prove that *CsTCP14* could transcriptionally repress the expression of the *CsNBS-LRR* gene, *Csa6M344280.1*, which resulted in inducing susceptibility to foliage diseases in cucumber. As such, we constructed a draft model showing that the leaf-specific expressed gene *CsTCP14* was negatively regulating the defense gene *Csa6M344280.1* to induce susceptibility to foliage diseases in cucumber. Therefore, this study explored key susceptible genes in response to foliage diseases based on a comprehensive analysis of public transcriptome data and provided an opportunity to breed new varieties that can resist foliage diseases in cucumber, as well as in other crops.

## 1. Introduction

Foliage diseases, including downy mildew (DM), target leaf spots, watermelon mosaic virus (WMV), and zucchini yellow mosaic virus (ZYMV), are extremely prevalent in cucurbit-crop production, causing serious yield reduction every year. Recently, several key resistance genes of foliage diseases in cucumber have been fine mapped [1,2,3], and its expression regulation model has been studied [4,5,6]. However, few researchers have paid attention to genes susceptible to plant disease, one of the hot topics in genetic research [7,8]. This is because susceptible genes are useful to apply to gene editing or RNA interference technology in breeding resistance varieties to defend against foliage diseases [7,8,9,10]. Several key susceptible genes of tomato, cucumber, and potato have been well explored and validated, such as the susceptible *MLO* and *ALS* genes to powdery mildew, and six S-genes to potato late blight [7,8,9]. Therefore, it is necessary to study susceptible genes in response to foliage disease in cucumber, which has been extensively cultivated all over the world (2.1 million hectares and 80.6 million tons, Food and agriculture organization (FAO), 2016).

It is well known that the TCP transcription factor is a key regulator of leaf morphogenesis in plants [11,12]. In *Arabidopsis*, simultaneous down-regulation of *TCP2*, *TCP3*, *TCP4*, *TCP10*, and *TCP24* resulted in large and crinkly leaves, whereas loss-of-function mutants of the individual *TCP* gene had a slight effect on leaf formation [13,14,15,16,17,18]. Conversely, hyper-activated *TCP4* resulted in the formation of smaller leaves. In *Arabidopsis*, the Class II TCPs were prior to Class I TCPs in regulating leaf formation, and they took functions in an antagonistic way [11]. In contrast to Class II TCPs above, the Class I *TCP20* mutant did not show any obvious phenotypic alterations but showed an increase in leaf pavement cell sizes under electron microscopy observation [11,19,20]. Recently, one study reported that a cucumber *TCP* gene, *TEN*, was controlling the formation of the tendril, which was an abnormal leaf and a combination of stems and leaves [21]. These results inferred that the TCP transcription factor genes may play a critical role in leaf formation and potentially be responsive to foliage diseases in cucumber.

The *TCP* genes were reported as being involved in regulating plant immune response. Firstly, *TCP* genes could target various pathogenic effectors [22,23,24,25]. Secondly, TCP TFs participate in the regulation of JA synthesis, which are key members in plant immune response [11,26,27,28,29]. In *Arabidopsis thaliana*, TCP proteins (TCP13, TCP14, TCP15, TCP19, and TCP21) can interact with three pathogen effectors, and a lack of *TCP13*, *TCP14*, and *TCP19* exhibited reduced immunity to pathogens [26,27]. Moreover, *TCP* genes were reported to be connectors in various signaling pathways, and six Class I TCP proteins were identified interacting with a negative regulator of effector-triggered immunity called SRFR1, which encodes an adaptor protein. Therefore, the TCP transcription factor genes were validated as being among the important regulators in plant resistance [29]. Further research is needed on whether *TCP* genes are involved in response to foliage diseases in cucumber.

Given the importance of TCPs in leaf development and plant immune response, in this study, a comprehensive expression analysis of the cucumber *CsTCP* genes was carried out based on public transcriptome data, to explore key genes involved in responding to foliage diseases in susceptible cucumbers. The leaf-specific expressed *CsTCP* genes were identified, and the expression pattern of genome-wide *CsTCP* genes under DM and WMV stresses was investigated. We identified a key susceptible TCP transcription factor gene *CsTCP14*, which was specifically responsive to two foliage diseases. Moreover, one of the important target genes of *CsTCP14* was explored by a Pearson correlation analysis of genome-wide defense gene expression. The obtained *CsNBS-LRR* gene, *Csa6M344280.1*, was transcriptionally repressed by *CsTCP14* after inoculation of DM and WMV, which may result in susceptibility to foliage diseases in cucumber.

## 2. Results

### 2.1. Identification of CsTCP Genes in Cucumber Genome

It is known that TCP genes are derived from the initials of TEOSINTE BRANCHED 1 (TB1) in Zea mays, CYCLOIDEA (CYC) in Antirrhinum majus, and PROLIFERATING CELL FACTORS 1 and PROLIFERATING CELL FACTORS 2 (PCF1 and PCF2) in Oryza sativa [30,31,32]. The genome-wide *CsTCP* genes were identified by Hidden Markov Models (HMM) search using TCP domain (PF03634) as a query. There were 27 CsTCP genes obtained and further confirmed by SMART and Interpro search, and localized on chromosome 1, 3, 4, 5, and 6 (Appendix A, Appendix A). The CsTCP27 was the longest gene, encoding 449 amino acids, whereas CsTCP8 was the shortest, encoding 174 amino acids. Multiple sequence alignment of CsTCPs was performed to reveal phylogenetic relationships (Appendix A). The CsTCP proteins contained the conserved basic helix-loop-helix (bHLH) structure and the R domain, of which the bHLH domain is proved to be responsible for DNA binding, nuclear targeting, and protein–protein interactions mediation [33,34], and the R domain is present in seven CsTCP proteins and may participate in protein-protein interactions through developing a hydrophilic α-helix, in accordance with reference [35].

All 27 CsTCPs were classified into two classes, Class I (used as Class 1 hereinafter) and Class II. Class I comprised 12 CsTCPs (44.4%), and Class II contained 15 CsTCPs (55.6%). Moreover, Class II TCPs can be further divided into two subclasses: six CsTCPs in the CYC/TB1 subclass and nine CsTCPs in the CIN subclass (Appendix A). This is consistent with previous reports. Class I and Class II *TCP* genes function in promoting plant development or inhibiting cell development, respectively, through activating a different function gene [33,36,37,38]. For convenience, the two subclasses were renamed as Class 2 (CYC/TB1) and Class 3 (CIN) (referred to as such hereinafter). Twenty-six of the 27 CsTCPs contained the conserved domain (bHLH) (Appendix A), and some Class 2 and Class 3 CsTCPs contained the conserved R domain “XX[E/V]XRX[K/R]-ARXRARXR[A/T]XX[E/K]”. Intriguingly, we identified a conserved amino acids sequence “I[A/E]ATGTGT[I/V]P[A/S]”, which was the first reported, hence, its function was still unknown (Appendix A). In addition, a phylogenetic tree based on 27 CsTCPs and 24 AtTCPs showed that they were clearly divided into three classes, and this result was consistent with the previous report in *Arabidopsis* (Figure 1) [39].

### 2.2. Identification of Leaf-Specific Expressed CsTCP Genes

The expression of *CsTCP* genes was obtained by a standard transcriptome analysis procedure based on a public dataset of RNA-seq in ten cucumber tissues, including root, stem, leaf, male flower, female flower, tendril, ovary, expanded ovary fertilized, and expanded ovary. (Figure 2) [40]. We found that 18 of the 25 detected *CsTCPs* specifically expressed in different tissues of cucumber. Among them, eight *CsTCPs* were highly expressed in leaves, including *CsTCP5*, *CsTCP13*, *CsTCP14*, *CsTCP15*, *CsTCP16*, *CsTCP22*, *CsTCP25*, and *CsTCP27* (Figure 2). Because *CsTCP5*, *CsTCP13*, *CsTCP15*, *CsTCP16*, and *CsTCP22*, were showing higher expression in other tissues, such as stem, ovary, flower, and tendril, *CsTCP14*, *CsTCP25*, and *CsTCP27* were considered as leaf-specific expressed genes in cucumber, and they may play critical roles in leaf morphological changes and potential involvement in resistance or susceptibility to foliage diseases (Figure 2).

### 2.3. Responsive Analysis of CsTCP Genes under Downy Mildew Stress

In this study, a comprehensive expression pattern of *CsTCP* genes was analyzed based on a public transcriptome of the susceptible cucumber line Vlaspik under inoculation of downy mildew (Figure 3) [41].

To investigate the potential functions of *CsTCPs* in resistance to DM, we performed *CsTCPs* expression model analysis after DM inoculation (Figure 3). We identified that ten *CsTCP* genes of the 14 detected *CsTCPs* were differently expressed compared with the control, during the first day to the eighth day after inoculation of downy mildew, *Pseudoperonospora cubensis* in cucumber leaves (Figure 3). The *CsTCP1*, *CsTCP12*, *CsTCP14*, *CsTCP15*, *CsTCP23*, and *CsTCP27* were observed continuously upregulating from 2 to 8 dpi (days post inoculation), indicating that they were induced to play roles under DM stress, whereas *CsTCP13* and *CsTCP25* were increasingly expressed on specific days after DM treatment, 2 dpi and 2 to 3 dpi (Figure 3). This result showed that these *CsTCP* genes were involved in susceptibility to downy mildew in cucumber, and *CsTCP14* and *CsTCP27* were important regulators in foliage diseases in DM because they were observed as leaf-specific genes in cucumber (Figure 2 and Figure 3).

### 2.4. Responsive Analysis of CsTCP Genes after Treatment with Watermelon Mosaic Virus

To confirm the responsive expression of *CsTCP* genes under foliage disease stress, especially in *CsTCP14*, *CsTCP25,* and *CsTCP27*, a watermelon mosaic virus treatment was applied within a susceptible cucumber line, Europe 8, and then the genome-wide *CsTCPs* expression was investigated using qRT-PCR technology (Figure 4). We found that all the *CsTCPs* were differentially expressed after WMV treatment, and eight *CsTCPs* were continuously upregulated with different expressed levels, such as *CsTCP10*, *CsTCP11*, *CsTCP12*, *CsTCP14*, *CsTCP17*, *CsTCP19*, *CsTCP23*, and *CsTCP25*. Among them, *CsTCP11*, *CsTCP12*, and *CsTCP14* had the most inducible expression from the first day to the 18th day after inoculation of WMV (Figure 4). Because *CsTCP14*, *CsTCP25*, and *CsTCP27* were considered leaf-specific expressed genes, we presumed that *CsTCP14* and *CsTCP25* were key regulators in susceptibility to WMV in cucumber.

### 2.5. Identification of CsTCP14 Inducing Susceptibility to Foliage Disease

Given that *CsTCP14*, *CsTCP25*, and *CsTCP27* were observed as leaf-specific expression genes in cucumbers (Figure 2), and *CsTCP14* and *CsTCP27* were increasingly expressed after downy mildew stress in susceptible cucumber line Vlaspik (Figure 3), and *CsTCP14* and *CsTCP25* were upregulated under WMV treatment in the susceptible cucumber Europe 8 (Figure 4), *CsTCP14* was both expressed specifically in leaves of cucumber and induced expression under foliage diseases DM and WMV stress in susceptible cucumbers. Therefore, the higher expression of *CsTCP14* was in accordance with foliage disease expansion in susceptible cucumbers, and it was assumed to be the most potentially critical regulator in inducing susceptibility to foliage disease in cucumber. This result was consistent with the study that suggested that *CsTCP14* homolog *AtTCP4* played key roles both in leaf development and in plant immune response in *Arabidopsis* [18].

### 2.6. Subcellular Localization of Susceptible Gene CsTCP14

In this study, two methods were applied to the localization of the key regulator *CsTCP14* involved in susceptibility to foliage disease. Firstly, an in-silico prediction of *CsTCP14* was localized in the nucleus based on an online search (Appendix A). Secondly, a transformation system was performed to validate the subcellular localization of *CsTCP14*. The pYBA1332::*CsTCP14*-GFP plasmid was constructed and bombarded into onion (*Allium cepa*) epidermal cells (Figure 5A), and the fluorescence microscopy revealed that the *CsTCP14*-GFP fusion protein was localized in the nucleus and cytoplasm (Figure 5A). This result was consistent with a previous study which confirmed that *FvTCP7*-GFP and *FvTCP17*-GFP were localized in both the nucleus and cytoplasm of *Arabidopsis* mesophyll protoplasts [42].

### 2.7. Discovery of Co-Expressed Defense Genes with the Susceptible Inducer CsTCP14

A Pearson correlation analysis between the expression of total *CsTCPs* and of the typical defense genes *CsNBS-LRRs* was conducted based on the transcriptome dataset of downy mildew inoculation, to discover the potential target genes of *CsTCP14*, whether they were positively or negatively involved in susceptibility to foliage disease (Appendix A). The Reads Per Kilobase per Million mapped reads (RPKM) expression values of 27 *CsTCPs* and 48 *CsNBS-LRR* genes were used to calculate the correlation coefficient, nine *CsNBS-LRRs* were positively co-expressed with *CsTCP14*, and another *CsNBS-LRR* gene *Csa6M133280.1* was significantly negatively co-expressed with *CsTCP14* (Appendix A). This result indicated that *CsTCP14* may transcriptionally regulate these defense genes by binding with their promoters. To validate this hypothesis, the cis-acting elements of these ten *CsNBS-LRR* genes were further investigated, and four *CsNBS-LRR* genes were assumed to be target genes of *CsTCPs* due to harboring TCP transcription factor specific binding sites ‘GGNCCC’, and the significant reducing gene *Csa6M344280.1* was one of the key targets, because there were three potential binding sites in its promoter (Appendix A). Given that the *CsTCP14* homolog *AtTCP4* in *Arabidopsis* played an important role in transcription inhibition [33,38], *CsTCP14* may target *CsNBS-LRR* in a transcriptional repression way, resulting in susceptibility to foliage disease in cucumber.

### 2.8. Repression Validation of CsTCP14 with Target Defense Gene CsNBS-LRR

In our study, an electrophoretic mobility shift assay (EMSA) test was carried out in vitro between *CsTCP14* and its potential target *CsNBS-LRR* gene, *Csa6M344280.1*. The Glutathione S-transferase (GST) -tagged CsTCP14 fusion protein was confirmed to bind to the element ‘GGNCCC’ in the *CsNBS-LRR* gene promoter (Figure 5B and Appendix A). The shifted band was attenuated gradually by increasing the concentration of unlabeled cold probes. The shifted band appeared when the core binding site “GGNCCC” was mutated into “GGNAAA”. Figure 5B shows that the *Csa6M344280.1* gene was a direct target of *CsTCP14*, and the binding site ‘GGNCCC’ was essential for CsTCP14 binding to the *CsNBS-LRR* gene. To further confirm this transcriptional repression, a yeast one-hybrid assay (Y1H) test was performed in vivo (Figure 5C). Only the Y1HGold-carrying PGADT7-CsTCP14 vectors and pBait-AbAi vectors could be grown in the SD/-Leu/AbA 100 medium, and the positive clones decreased gradually with an increase of dilution concentration. This result proved that pGADT7-TCP14 encoded protein specifically bound to the GGTCCC region of the *CsNBS-LRR* (*Csa6M344280.1*) promoter. Therefore, the EMSA and Y1H experiments validated that the susceptible regulator *CsTCP14* could repress the defense gene *Csa6M344280.1*, resulting in susceptibility to foliage disease in cucumber.

## 3. Discussion

### 3.1. Identification of Leaf-Specific Expressed CsTCPs in Regulating Leaf Morphogenesis

It is well known that gene expression is correlated with its function [43]. This study analyzed the expression patterns of all 27 *CsTCP* genes in ten different tissues of cucumber, including leaf, stem, ovary, flower, and tendril. We found that *CsTCP14*, *CsTCP25,* and *CsTCP27* were observed specifically expressed in leaves (Figure 2). As previously reported, the close genetic homologous genes of *CsTCP14*, *CsTCP25*, and *CsTCP27* in *Arabidopsis* were *AtTCP3*, *AtTCP4*, and *AtTCP24*, respectively, and they were well conferred to regulate leaf development [13,15,18]. Additionally, *CsTCP14* and *CsTCP21* were observed as highly expressed in tendril base and in tendril, which was recognized as an abnormal leaf in cucumber, and *CsTCP21* was reported as *TEN* in controlling tendril formation [21]. Moreover, the *AtTCP4* and *AtTCP20* were proved to regulate leaf development via Jasmonic acid (JA) pathway antagonistically in *Arabidopsis*, and their homologous genes, *CsTCP14* and *CsTCP11*, were assumed to be key regulators of leaf development, maybe via the JA pathway [11]. Therefore, our results identified three key leaf-specific expressed genes (*CsTCP14*, *CsTCP25*, and *CsTCP27*) in cucumber, and they may play critical roles in leaf morphogenesis, and be potentially involved in the response to foliage diseases in cucumber.

### 3.2. Identification of Susceptible CsTCP Genes under Foliage Disease Stress

The TCP TFs were reported to participate in plant resistance response through different immune processes [22,23,24,25]. This study investigated the expression of the total *CsTCPs* in response to foliage diseases, which was in accordance with previous results showing that the *TCP* genes could participate in physiological processes of plant immunity [29]. After inoculation of DM in susceptible cucumber, six *CsTCPs* were observed as upregulating, including *CsTCP1*, *CsTCP12*, *CsTCP14*, *CsTCP15*, *CsTCP23*, and *CsTCP27* (Figure 3). In addition, in treatment of WMV in susceptible cucumber, eight *CsTCPs* were upregulated at a high level; they were *CsTCP10*, *CsTCP11*, *CsTCP12*, *CsTCP14*, *CsTCP17*, *CsTCP19*, *CsTCP23*, and *CsTCP25* (Figure 4). Combining with the expression of *CsTCP14*, *CsTCP25* and *CsTCP27* were observed specifically expressed in leaf tissues (Figure 2), and *CsTCP14* was investigated as inducible responder under foliage disease stresses, as well as in regulating leaf morphogenesis in cucumber, especially in susceptible cucumber. Given that the *CsTCP14* homolog *AtTCP4* was reported as a key player both in leaf development and in plant immune response in *Arabidopsis* [13,15,18], *CsTCP14* was further identified as the most important gene to induce susceptibility to foliage diseases in cucumber.

### 3.3. Transcriptional Repression Regulatory of CsTCP14 and Defense Gene

In this study, the expression of 48 defense gene *CsNBS-LRRs* were investigated and analyzed with the key susceptible *CsTCP14* in cucumber, and the *CsNBS-LRR* gene *Csa6M133280.1* negatively co-expressed with *CsTCP14* significantly (Appendix A). Moreover, the promoter of *Csa6M344280.1* was observed existing in three potential binding sites of ‘GGNCCC’, which was the key binding site of TCPs transcription factors in plants (Appendix A). Given that the cucumber line tested in this study and the genome reference line 9930 were susceptible to DM and WMV, and the defense gene *Csa6M344280.1* was downregulated under foliage diseases, it was considered an extremely curtailing regulator in inducing susceptibility to foliage diseases. Then, the transcriptional regulatory mechanism was applied through EMSA and Y1H tests, and the result confirmed that *CsTCP14* could repress the expression of *Csa6M344280.1* by binding its promoter region. As such, we constructed one possible model of *CsTCP14* inducing susceptibility to foliage diseases by transcriptionally repressing the defense gene *CsNBS-LRR* in cucumber (Figure 6). This result was in accordance with previous reports that the *CsTCP14* homolog *AtTCP4* in *Arabidopsis* played an important role in transcription inhibition [33,38]. Therefore, this study established a comprehensive analysis of exploring key susceptible genes in response to foliage diseases and validated the transcriptional regulation of these genes in cucumber. Our new finding would be useful for gene edition technology in breeding new varieties resistant to foliage diseases in cucumber, as well as their homologs in other crops.

## 4. Materials and Methods

### 4.1. Identification and Chromosomal Localization of CsTCP Genes

To identify cucumber *TCP* genes [44], the genome sequence of cucumber and the HMM profile of the TCP domain (PF03634) were downloaded from database CuGenDB (http://cucurbitgenomics.org/) and Pfam protein family database (http://pfam.xfam.org/), respectively. The candidate *TCP* genes were further inspected with SMART search (http://smart.embl-heidelberg.de) and Interpro analysis (http://www.ebi.ac.uk/interpro/scan.html) [45,46]. The chromosomal localization of *CsTCP* genes was retrieved from *C. sativus* genome 9930 V2.0. The *CsTCP* genes were named from *CsTCP1* to *CsTCP27* according to their chromosomal locations, and the isoelectric point and molecular weight of CsTCPs were estimated by ExPASy server (http://web.expasy.org/compute_pi) [47]. The subcellular localization of CsTCP proteins was predicted using an online tool CELLO (http://cello.life.nctu.edu.tw/) [48].

Phylogenetic analysis was conducted by Cluster W software (Version 7), which was employed to perform protein multiple sequence alignments of 27 CsTCPs. The alignment was subsequently imported into MEGA6.0 software to construct an unrooted phylogenetic tree by the Neighbor-Joining (NJ) method with 1,000 bootstrap replicates [49]. The amino acid sequences of TCPs in *Arabidopsis thaliana* were obtained from previous studies [35]. The TCP evolution between cucumber and *Arabidopsis* was studied through an unrooted phylogenetic tree, acquired from the same method above.

### 4.2. Transcriptome Analysis of Leaf-Specific Expressed CsTCPs in Cucumber

The tissue-specific expression patterns of *CsTCPs* were analyzed based on the published RNA-seq data with an accession number of SRA046916 [40]. The transcriptome profile analysis was on 10 different cucumber tissues, including root, stem, leaf, male flower, female flower, ovary, expanded ovary, expanded ovary fertilized, and tendril. The RPKM value was recalculated by a standard transcriptome analysis protocol [6]. The genome-wide expression of *CsTCP* genes was shown on a heatmap using HemI 1.0 software, and the higher expression was indicated by a color bar changing from black to red.

### 4.3. Transcriptome Analysis of CsTCPs Regulatory after Inoculation of Downy Mildew

The expression regulation of *CsTCP* genes responsive to downy mildew stress was retrieved from a public transcriptome dataset of RNA-seq, which was conducted for revealing the genome-wide differential expressed genes after inoculation with *Pseudoperonospora cubensis* (SRP009350) [41]. The susceptible cucumber line Vlaspik was treated with downy mildew on leaves, and the leaf sample was investigated at 0, 1, 2, 3, 6, and 8 days after inoculation (dpi). Because the transcriptome was sequenced by single-end technology, the RPKM value was recalculated by a standard transcriptome analysis protocol according to a previous report [6], and the expression of *CsTCP* genes was shown by a heatmap using HemI 1.0 software in this study.

### 4.4. Expression Analysis of CsTCPs Responsive to WMV Treatment

#### 4.4.1. Materials and WMV Inoculation

The cucumber line Europe 8, which is susceptible to WMV, were cultivated in a greenhouse (Beijing Vegetable Research Center). The WMV was diluted to a concentration of 1:3 (*w*/*v*) in 0.2 mol·L^−1^ phosphoric acid buffer (pH 7.0) when the second true leaf was fully expanded. The leaves were inoculated with a small amount of 600 to 800 mesh emery after rinsing with clear water, and a control setting was inoculated with phosphoric acid buffer (three duplications and 20 seedlings for each treatment). Both the inoculated and control seedlings were cultivated under the correct conditions (insect-free, ~30 °C, 90% relative humidity).

#### 4.4.2. RNA Extraction

Leaf samples were collected from 3 independent plants in each treatment of the 0, 1, 3, 6, 9, 12, 18, and 24 days post-inoculation (dpi), and they were immediately frozen in liquid nitrogen. Total RNA was isolated with Quick RNA isolation Kit (Huayueyang, Beijing, China) following the manufacturer’s recommended protocol. An aliquot of total RNA was treated with RNase-free DNase I (NEB, Ipswich, MA, USA) to degrade contaminating genomic DNA. The qualities and quantities of total RNA were assessed on NanoDrop ND-2000 Spectrophotometer (Thermo Fisher Scientific Inc., Waltham, MA, USA). The total RNA was subsequently reverse transcribed to double-stranded cDNA using Super Script III Reverse Transcriptase (TaKaRa, Dalian, China).

#### 4.4.3. Real-Time Quantitative PCR

The specific primer design of each *CsTCP* gene was implemented on an online Primer3 website (http://primer3.ut.ee), and are shown in Appendix A. The reference gene sequences that quantitative real-time (qRT-PCR) used are 5′-GGCAGTGGTGGTGAACATG-3′ and 5′-TTCTGGTGATGGTGTGAGTC-3′. The qRT-PCR setting to three replications was performed using an SYBR Premix Ex TaqTM kit (TaKaRa, Dalian, China) and a Roche LightCycler480 System (Bio-Rad, Hercules, CA, USA). Relative expression levels of *CsTCPs* were calculated with the comparative threshold method (2^−ΔΔ*C*t^) and shown on a heatmap using HemI 1.0 software. The expression of *CsTCPs* under WMV stress was analyzed by real-time quantitative PCR (qRT-PCR) technology; treatments covered 1, 3, 6, 9, 12, and 18 days after inoculation.

### 4.5. Identification of Potential Target Genes of CsTCPs’ Susceptibility to Foliage Disease

The TCP genes were reported as being involved in regulating plant immune response by transcriptional regulation of their target genes [24]. In this study, the potential target genes of *CsTCPs* involved in resistance or susceptibility to foliage disease were explored. A Pearson correlation analysis was conducted to investigate the correlation between the key *CsTCPs* and their potential targets. This study focused on the typical defense genes, *CsNBS-LRRs*, in cucumber, which was a key participator in response to disease. The transcriptome profile dataset under downy mildew was used, and the correlation coefficient and significance were calculated in the R psych packages, the significance correlation (*p* < 0.05) between *CsTCP* genes and *CsNBS-LRRs* was indicated by a star (*), and extreme significance correlation (*p* < 0.01) was indicated by two stars (**) in this result.

### 4.6. Transactivation Repression Validation of CsTCP14 and Its Target Genes

#### 4.6.1. Subcellular Localization of *CsTCP14*

The *CsTCP14* (*Csa4M628330.1*) coding sequence was amplified with primers listed in Appendix A. Subsequently, *CsTCP14* was inserted into the pYBA1332-GFP vector. Both the pYBA1332::GFP and pYBA1332::*CsTCP14*-GFP vectors were bombarded into onion bulb scale epidermal cells. After incubation in the dark at 25 °C for 6 h, fluorescence signals were detected using a Laser Scanning Confocal Microscope (Nikon A1R, Tokyo, Japan).

#### 4.6.2. Expression of CsTCP14 Protein

The sequence of the *CsTCP14* gene was amplified from cucumber 9930 with specific primers (Appendix A) and cloned into the pGEX4T-1 vector carrying GST tag. The pGEX4T-1-*CsTCP14* was transformed into *Escherichia coli* strain BL21 (DE3) for further culture. The 0.2 mM Isopropyl-β-D-thiogalactopyranoside (IPTG) was added to the bacteria culture when the concentration reached 0.6 at 600 nm (OD600). The recombinant GST-CsTCP14 protein was harvested after culture at 37 °C for 5 h with 200 rpm. To confirm the correctness of the CsTCP14 expression, Western blot was performed to identify the fusion protein with the right Odyssey CLx Imaging System (LI-COR, Lincoln, NE, USA). The GST-CsTCP protein was purified by Bio-Scale Mini Profinity GST cartridge (Bio-Rad, CA, USA) and size exclusion column (Bio-Rad, Hercules, CA, USA).

#### 4.6.3. Electrophoretic Mobility Shift Assay

The purified protein was subjected to a Light Shift Chemiluminescent EMSA kit (Thermo, Waltham, MA, USA). The *CsNBS-LRR* gene *CSa6M344280.1* probe and its mutation probes (Appendix A) were selected to identify the authenticity of the binding site. Among them, the *CsNBS-LRR* probes without mutation were 5′-biotin-labeled. Meanwhile, the same sequences with non-labeled biotin were used as the competitive probes for EMSA. The 20 μL binding reactions were electrophoresed in 6.5% native polyacrylamide gel, and then electro transferred onto a nylon membrane. The membrane was detected using a ChemiDocTM MP Imaging System (Bio-Rad, CA, USA).

#### 4.6.4. Yeast One-Hybrid Assay

Yeast one-hybrid assay was conducted to validate the transcriptional regulation of *CsTCP14* and its targets based on the protocol in Clontech’s Matchmaker Gold Yeast One-Hybrid System (Clontech Laboratories, Inc., a Takara Bio Company, Mountain View, CA, USA). The pGADT7 prey vector carrying *CsTCP14* was constructed using the primers pGADT7-F and pGADT7-R (Appendix A). The *CsNBS-LRR* promoter fragment containing the TCP binding site “GGNCCC” was shown as the bait sequence. To obtain genuine positive clones, we also mutated the TCP binding site “GGNCCC” to “GGTAAA” (designed as Bait-m). The Bait and Bait-m sequences were cloned into the pAbAi vector by using the primers listed in Appendix A. The linearized pBait-AbAi and pBait-m-AbAi plasmids with BstB I were cloned into yeast strain Y1HGold. The transformed yeast cells carrying an empty pGADT7 vector were set up as negative controls. The Y1HGold were cultured in the optimal concentration of AbA. The experiments were repeated three times.

### 4.7. Statistical Analysis

A one-way analysis of variance (ANOVA) test was employed to analyze the variance and significant difference among *CsTCP* expressions under different conditions. The software used for the statistical analyses was SPSS 16.0 (SPSS Company, Chicago, IL, USA).

## 5. Conclusions

In this study, we identified a key regulatory gene, *CsTCP14*, which was responsive to susceptibility to foliage disease, downy mildew, and watermelon mosaic virus in cucumber. The *CsTCP14* was observed as a leaf-specific expressed gene and was inducible to upregulation under downy mildew and watermelon mosaic virus treatment. Further, the potential target defense gene, *Csa6M344280.1*, of *CsTCP14* under foliage disease stress was explored and validated by EMSA and Y1H experiment. This study constructed a draft model to demonstrate that the leaf-specific expressed gene *CsTCP14* was negatively regulating the defense gene *Csa6M344280.1* to induce susceptibility to foliage diseases in cucumber.

## Figures and Tables

**Figure 1 ijms-20-02582-f001:**
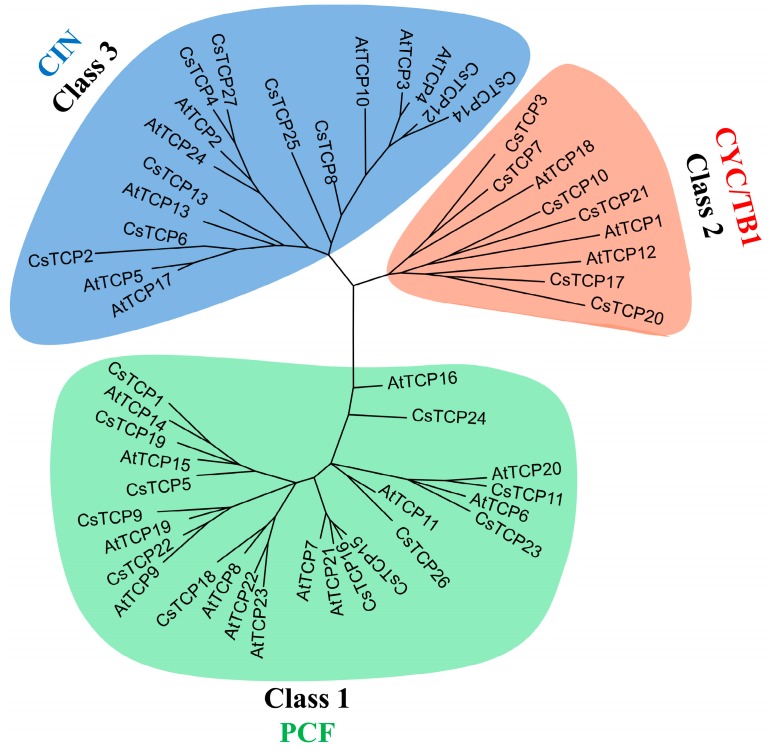
The phylogenetic tree of the total CsTCPs and AtTCPs. It was constructed in the MEGA 6.0 software with the Neighbor-Joining method. The bootstrap analysis was performed with 1000 iterations. Different colors represented different TCP classes, Class 1 (PCF), Class 2 (CYC/TB1) and Class 3 (CIN).

**Figure 2 ijms-20-02582-f002:**
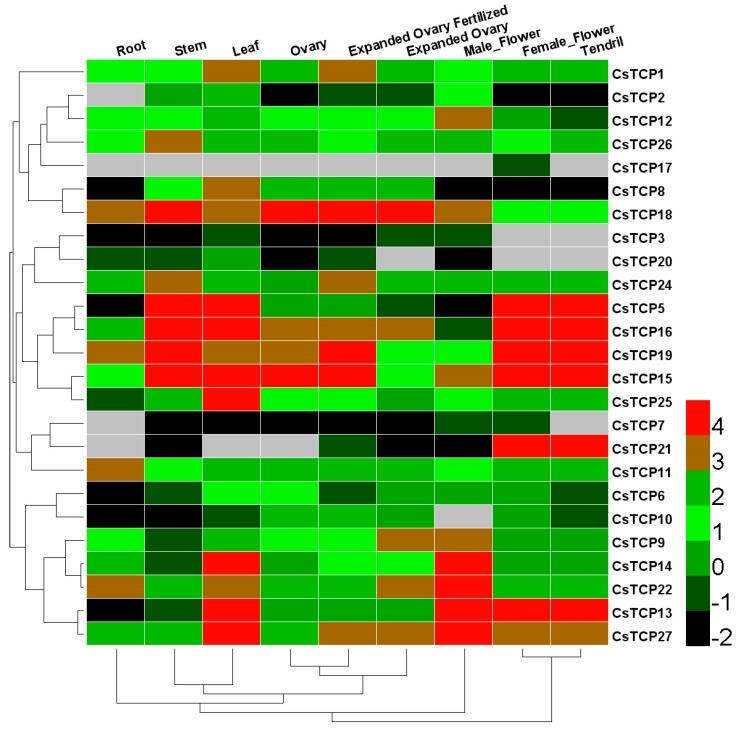
Tissue-specific expression of *CsTCP* genes in cucumber. The *CsTCP* genes transcripts in nine tissues of cucumber 9930 were determined by public transcriptome data. The color scale showed that the increasing expression levels form black to red.

**Figure 3 ijms-20-02582-f003:**
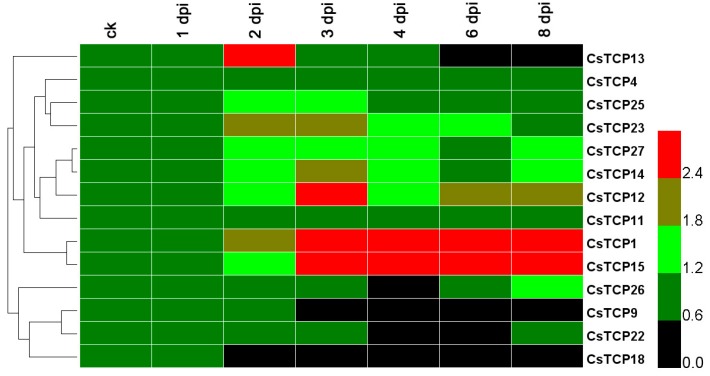
Responsive analysis of *CsTCP* genes under downy mildew stress. The *CsTCP* genes transcripts were determined from day post inoculation 1 to 8 after DM treatment, compared with in the control (CK) without inoculation. The color scale shows that the increasing expression levels form black to red.

**Figure 4 ijms-20-02582-f004:**
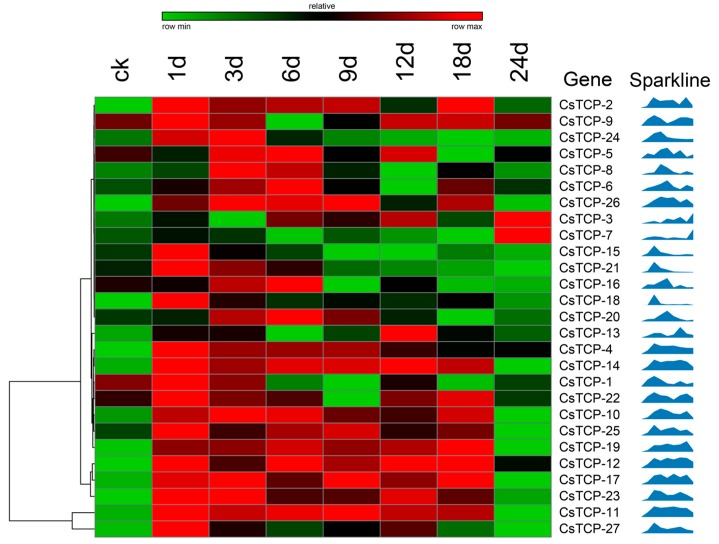
Responsive expression of CsTCP genes under WMV stress. The gene expression profiles were investigated through qRT-PCR in day 1, 3, 6, 9, 12, 18, and 24 under WMV stress, compared with in CK without WMV treatment. The color scale showed that the increasing expression levels form black to red on the top, and the sparkline was shown on the right.

**Figure 5 ijms-20-02582-f005:**
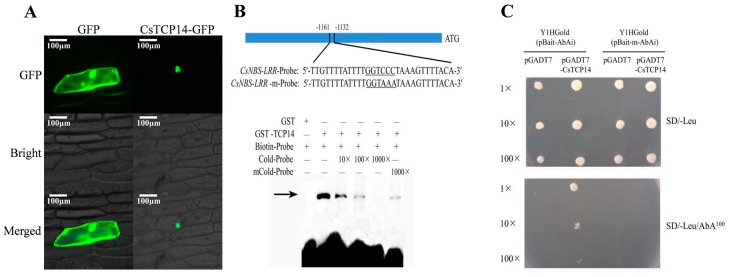
Transcriptional repression of CsTCP14 with defense gene *CsNBS-LRR* (*Cs6M344280.1*). (**A**) The pYBA1332::CsTCP14-GFP was localized in nuclear of the onion epidermal cell, indicated as CsTCP14-GFP. The empty vector of pYBA1332::GFP was set up as control, indicated as GFP. The scale bar was 100 μm. (**B**) The EMSA experiment showed that the GST-TCP14 fusion protein was attenuated gradually by increasing the concentration of unlabeled cold-probes, while as it was reappeared after incubated with the mutated cold-probes. The band was indicated by the black arrow, and the experiment was performed in three replications. (**C**) The Y1H experiments showed that the positive clones were decreasing gradually when increasing the dilution concentration of pGADT7-CsTCP14. The empty vector pGADT7 was set up as negative controls, and three replications were performed.

**Figure 6 ijms-20-02582-f006:**
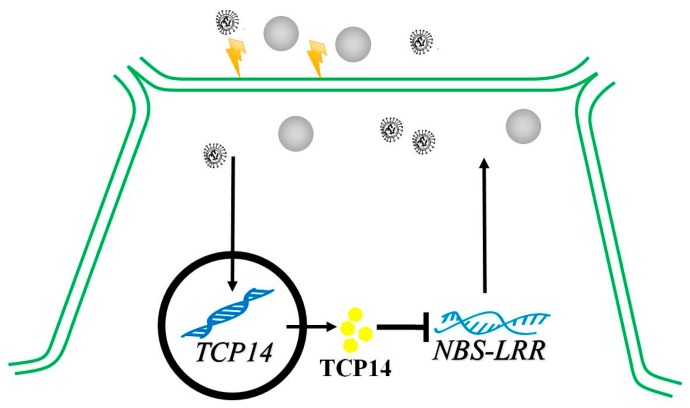
The deducted model of CsTCP14 regulatory the susceptibility to foliage diseases. The *CsTCP14* was upregulated under foliage disease attack, then repressing the expression of defense gene *CsNBS-LRR* transcriptionally, induced the susceptibility to foliage disease.

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
