# Peer review of "Transcriptome Profile Analysis Reveals that CsTCP14 Induces Susceptibility to Foliage Diseases in Cucumber"

_ijms, 2019, doi:10.3390/ijms20102582_

Round 1

Reviewer 1 Report

The paper is a transcriptome study focused on discovery of regulatory genes, responsible for susceptibility of cucumbers to foliage diseases. Moreover, the study identifies the potential target gene of this regulatory gene and highlights the mechanism of its inhibition. The paper is very well structured and interesting for reading. The experiments are well structured and performed in the compliance with the latest scientific standards in the field. The results are presented on clear and understandable way and fully support the conclusions made by the authors.

There are some minima technical points, which should be addressed in order to improve the presentation of the paper:

53 – format “Arabidopsis” in italic;

66-67 – format “Arabidopsis thaliana” in italic;

113 - format “Arabidopsis” in italic;

165 – “Gene name and color scale were shown at the right.” –here the color scale is on the top and here is “Sparkline” on the right. Please correct;

177-178 – “an exotic transformation system” – revise this;

183 - format “Arabidopsis” in italic;

198 - format “Arabidopsis” in italic;

209 – “SD/-Leu medium,” should be “SD/-Leu/AbA100 medium,”

232 - format “Arabidopsis” in italic;

237 - format “Arabidopsis” in italic;

269 - format “Arabidopsis” in italic;

285 – format “C. sativus” in italic;

295 - format “Arabidopsis” in italic;

349 – 350 – “the significance correlation” should be “the significance correlation (P<0.05)”;

350 – 351 – “extremly significance correlation” should be “extremely significance correlation (P<0.01)”

Author Response

Dear reviewer, thank you very much for your professional review, and we made a point-by-point revision based on your nice comments.

Dear reviewer, we made an extensive editing of language according the recommendation in MDPI system, please see the attachment evidence of editing.

Please check the point-by-point revision as follow:

1.    53 – format “Arabidopsis” in italic;

Response: Thanks, we revise it, and please check the line 55 in the new manuscript.

2.    66-67 – format “Arabidopsis thaliana” in italic;

Response: Revise as suggested. Please check the line 69 in the new manuscript.

3.    113 - format “Arabidopsis” in italic;

Response: We've modified it on line 113 of the new manuscript.

4.    165 – “Gene name and color scale were shown at the right.” –here the color scale is on the top and here is “Sparkline” on the right. Please correct;

Response: Thank you very much, we corrected it and rewrite the legend on line 171.

5.    177-178 – “an exotic transformation system” – revise this;

Response: We modified it as “a transformation system” on line 186-187.

6.    183 - format “Arabidopsis” in italic;

Response: We revised it on line 192 of the new manuscript.

7.    198 - format “Arabidopsis” in italic;

Response: Revise as suggested on line 207.

8.    209 – “SD/-Leu medium,” should be “SD/-Leu/AbA100 medium,”

Response: Revision made as reminded. Please check on line 220 of the new manuscript. Thanks.

9.    232 - format “Arabidopsis” in italic;

Response: Change made as suggested on line 243, Thank you.

10.   237 - format “Arabidopsis” in italic;

Response: Change made as suggested on line 248.

11.   269 - format “Arabidopsis” in italic;

Response: Typos corrected on line 281.

12.   285 – format “C. sativus” in italic;

Response: Error corrected, on line 298 of the new manuscript.

13.   295 - format “Arabidopsis” in italic;

Response: Typos corrected on line 308.

14.   349 – 350 – “the significance correlation” should be “the significance correlation (P<0.05)”;

Response: Change made as suggested. Please check on line 363.

15.   350 – 351 – “extremly significance correlation” should be “extremely significance correlation (P<0.01)”

Response: Typos corrected, and we add the (P<0.01) in the new manuscripts on line 364. Thanks.

Dear reviewer, we really appreciate for your professional review.

Have a nice day.

Best regards

Changlong Wen

Reviewer 2 Report

Review

In the work entitled “Transcriptome Profiles Analysis Reveals CsTCP14 Involves in Susceptibility to Foliage Diseases in Cucumber” the authors performed an in-silico analysis of TCP gene family in cucumber in order to unveil genes involved in foliage diseases responses. A further analysis revealed the putative action of CsTCP14 in targeting an CsNBS-LRR gene (Csa6M344280.1) in leaves during susceptibility to foliage diseases.

The quality of pictures is good and clear and illustrate well the results.

Major points to consider:

The manuscript requires a major English review. It contains numerous grammar errors making the reading quite difficult. Discussion needs particular attention. The authors ought to perform a deeper discussion specially when it comes to the mechanisms of action of the gene TCP14.

-          Figure 5 needs attention since the internal legend, especially in picture A, is impossible to read.

-          It is also necessary to clear out that the in-silico analyses were performed making use of public data and not obtained by the authors.

-          Please, stress out why the authors picked the CsTPC14 gene for further analysis. It is not very clear in the results.

-          For me it is not clear at all it the up-regulation of CsTPC14 increases or reduces the susceptibility of the plant to foliage diseases. Stating things like “involve in the susceptibility of foliage diseases in cucumber” does not mean anything. Does the plant become more or less susceptible to diseases with an up-regulation of TCP14? The definition of susceptibility is: “the state or fact of being likely or liable to be influenced or harmed by a particular thing”.

Author Response

Dear reviewer, thanks a lot for your professional review, and we made a point-by-point revision based on your kindly comments.

Dear reviewer, Firstly, we made an extensive editing of language according the recommendation in MDPI system, please see the attachment evidence of editing.

Secondly, the point-by-point revision was as follow:

1.       Thank you very much. We rewrite all of figure legends in the new manuscript, especially the legend of Figure 5 was revised as follow:

Figure 5. Transcriptional repression of CsTCP14 with defense gene CsNBS-LRR (Cs6M344280.1). (A) The pYBA1332::CsTCP14-GFP was localized in nuclear of the onion epidermal cell, indicated as CsTCP14-GFP. The empty vector of pYBA1332::GFP was set up as control, indicated as GFP. The scale bar was 100 μm. (B) The EMSA experiment showed that the GST-TCP14 fusion protein was attenuated gradually by increasing the concentration of unlabelled cold-probes, while as it was reappeared after incubated with the mutated cold-probes. The band was indicated by the black arrow, and the experiment was performed in three replications. (C) The Y1H experiments showed that the positive clones were decreasing gradually when increasing the dilution concentration of pGADT7-CsTCP14. The empty vector pGADT7 was set up as negative controls, and three replications were performed.

2.       Thanks for your kind remind, In the new manuscript, we noticed that we performed a public transcriptome data in the abstract (Line 22 and 36), introduction (Line 77), result (Line 121 and 136), figure legend (Line 132), discussion (Line 236), methods (Line 317). And we really thanks to these public resources supplying valuable information.

3.       Thanks a lot. In this study, we revise a new paragraph about exploring the key susceptible gene CsTCP14. Please check it as follow:

2.5. Identification of CsTCP14 inducing susceptibility to foliage disease

Given that CsTCP14, CsTCP25, and CsTCP27 were observed as leaf-specific expression genes in cucumbers (Figure 2), and CsTCP14 and CsTCP27 were increasingly expressed after downy mildew stress in susceptible cucumber line Vlaspik (Figure 3), and CsTCP14 and CsTCP25 were upregulated under WMV treatment in the susceptible cucumber Europe 8 (Figure 4), CsTCP14 was both expressed specifically in leaves of cucumber and induced expression under foliage diseases DM and WMV stress in susceptible cucumbers. Therefore, the higher expression of CsTCP14 was in accordance with foliage disease expansion in susceptible cucumbers, and it was assumed to be the most potentially critical regulator in inducing susceptibility to foliage disease in cucumber. This result was consistent with the study that suggested that CsTCP14 homologue AtTCP4 played key roles both in leaf development and in plant immune response in Arabidopsis [18].

4.       We appreciate for your advice, in this new version of manuscript, we emphasized that the CsTCP14 were key regulator to induce susceptibility to foliage diseases in the susceptible cucumber, and we revised this important point in the title (Line 3), abstract (Line 34), results (Line 143, 175, 191), discussion (Line 263) and conclusion (Line 410) of our manuscript. In the meantime, we observed the CsTCP14 could induces the susceptibility to another foliage disease, like the Papaya ringspot virus (PRSV, unpublished data). And we attempted to test the function of CsTCP14 by a transformation system in cucumber, which will be figuring out its function in next year.

Dear reviewer, we really appreciate for your professional review.

Have a nice day.

Best regards

Changlong Wen

Round 2

Reviewer 2 Report

In the work entitled “Transcriptome Profiles Analysis Reveals CsTCP14 Involves in Susceptibility to Foliage Diseases in Cucumber” the authors performed an in-silico analysis of TCP gene family in cucumber in order to unveil genes involved in foliage diseases responses. A further analysis revealed the putative action of CsTCP14 in targeting an CsNBS-LRR gene (Csa6M344280.1) in leaves during susceptibility to foliage diseases.

Thank you for addressing all the suggestions made earlier. The manuscript has improved considerably, and I have no further comments.